# SimGuard: Context-Aware Anomaly Filtering via Similarity-Guided Error Detection

**Amir Raza** [1]   **Mayank Jauhari** [1]   **Vikash Sharma** [1]   **Vipul Joshi** [1]
**Boris N Oreshkin** [2]   **Anurag Tripathi** [1]

## Abstract

Given millions of invoices flagged daily by enterprise anomaly detectors, how can we reliably separate genuine errors from benign anomalies without drowning human investigators, when a single missed error can cost millions and vendor-specific retraining is infeasible? We present **SimGuard**, a context-aware error-detection framework built on two ideas: *asymmetric temperature decoupling* in the InfoNCE loss, which prevents feature suppression on heterogeneous tabular data, and a *curriculum* over the denominator temperature $\tau_{\mathrm{denom}}$ and corruption rate $p_{\mathrm{corr}}$ that sharpens hard-negative discrimination without losing rare-error sensitivity. At inference, SimGuard retrieves the $k$ most similar historical cases within a vendor partition and filters anomalies that match known benign patterns via a similarity-weighted vote over investigator-verified labels. SimGuard operates as a post-hoc filtering layer downstream of foundation-model classifiers such as TabPFN (Hollmann et al., 2025), whose high recall (0.92) generates excessive false positives that domain-specific filtering must resolve. On enterprise-scale invoice datasets SimGuard reduces flagged anomalies by **15%** while keeping recall loss **below 1%**; paired with TabPFN on public fraud benchmarks it achieves Santander F1 of **0.53** (vs. TabPFN alone 0.51) and Credit Card Fraud F1 of **0.77 to 0.78** (vs. 0.73) at volume reductions of **9.4% to 31.0%**. Each SimGuard prediction comes with its retrieved neighbors as case-level explanations, and the system absorbs new error patterns by adding investigator-labeled cases to the repository, with no retraining required (full results in App. D).

[1]Amazon, India [2]Amazon, USA. Correspondence to: Amir Raza <amirraz@amazon.com>.

*Proceedings of the $2^{nd}$ ICML Workshop on Foundation Models for Structured Data*, Seoul, South Korea. 2026. Copyright 2026 by the author(s).

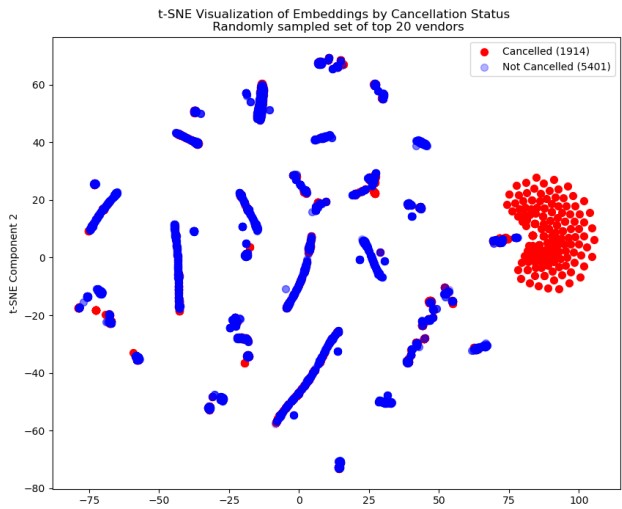

*Figure 1.* **SimGuard separates errors from benign anomalies.** t-SNE of SimGuard embeddings on the top-20 vendors: erroneous invoices (red) form distinct clusters from benign anomalies (blue). This geometric separation enables SimGuard's 15% volume reduction at <1% ΔRecall on enterprise data (Table 3).

## 1. Introduction

Given millions of invoices flagged annually by enterprise anomaly detectors, how can we filter out the benign anomalies so investigators spend their scarce time on genuine errors? This question is high-stakes: **a single missed error can cost millions**, while each false positive wastes expert time. Foundation-model classifiers such as TabPFN (Hollmann et al., 2025) offer strong out-of-the-box recall on tabular fraud tasks, but their precision remains low: on our enterprise data a TabPFN first stage achieves 0.92 recall at only 0.20 precision, flooding investigators with false positives. Context matters, since an anomaly at one vendor may be normal elsewhere, and error patterns evolve faster than supervised models can retrain (see App. A for extended discussion).

**SimGuard.** We introduce **SimGuard**, which (a) learns an **encoder** $\phi: \mathcal{X} \to S^{d-1}$ on heterogeneous tabular data via a SCARF-style contrastive objective with asymmetric temperature decoupling and a curriculum corruption schedule,

and (b) at inference retrieves the $k$ most similar historical cases $\{x_1, \ldots, x_k\} \subset \mathcal{H}$ within a vendor partition and computes a risk score $s(x_q) = f(\{y_1, \ldots, y_k\})$ from their investigator-verified labels $y_i \in \{0, 1\}$. Table 1 summarizes what SimGuard uniquely provides vs. representative baselines.

**Problem and positioning.** Given an anomaly-flagged invoice $x_q$ and a per-vendor history $\mathcal{H}$ of investigator-labeled cases, SimGuard predicts whether $x_q$ is a true error ($y{=}1$) or benign ($y{=}0$) with high recall and low review volume. The key technical move is asymmetric InfoNCE: standard contrastive objectives tie a single $\tau$ to both the alignment term (positive-pair invariance) and the dispersion term (negative-pair separation), forcing a trade-off; decoupling $\tau_{\text{num}}$ from $\tau_{\text{denom}}$ preserves rare-feature sensitivity while sharpening hard-negative discrimination. The gain concentrates when features mix categorical and numerical types and hard negatives are common; for extremely imbalanced or well-separated regimes (e.g., IEEE Fraud at 3.59%), tree baselines remain competitive (§4).

We present the following **contributions**:

1. **Asymmetry.** A novel asymmetric InfoNCE that decouples $\tau_{\text{num}}$ and $\tau_{\text{denom}}$, preventing feature suppression on tabular data (Thm. 4, App. E.5).

2. **Curriculum.** A 50-epoch alternating schedule over $(\tau_{\text{denom}}, p_{\text{corr}})$ that guides training from broad clustering to fine-grained discrimination (App. C.3).

3. **Adaptivity.** Vendor-scoped few-shot retrieval absorbs new error patterns without retraining and yields case-level explanations (Thm. 8, App. E.5).

Section 3 details the method; Section 4 evaluates SimGuard on enterprise invoices and three public fraud benchmarks.

## 2. Related Work

Classical anomaly detection (Chandola et al., 2009; Aggarwal, 2017) and deep methods (autoencoders (Zhou & Paffenroth, 2017), DeepSVDD (Ruff et al., 2018), DROCC (Goyal et al., 2020)) typically assume labeled anomalies or representative examples from base classes, constraints impractical in financial settings where error patterns continuously evolve. Deep learning on tabular data still trails gradient-boosted trees (Grinsztajn et al., 2022) despite specialized architectures like TabNet (Arik & Pfister, 2021) and TabPFN (Hollmann et al., 2025). Self-supervised contrastive learning (Chen et al., 2020; van den Oord et al., 2018) has closed this gap for some modalities; SCARF (Bahri et al., 2022) adapts it to tabular data via random feature corruption. Temperature scheduling (Kukleva et al., 2023) and decoupled-temperature variants (Dinu et al., 2025) have improved contrastive representations in vision and text. Robin-

son et al. (2021) showed that standard InfoNCE can admit feature-suppressing minimizers, motivating our asymmetric decoupling. Unlike Dinu et al. (2025), who apply decoupled temperatures to sentence embeddings for compression, SimGuard targets tabular data where feature suppression is especially pernicious due to categorical/numerical heterogeneity, pairing asymmetric decoupling with SCARF-style corruption and a curriculum schedule.

## 3. Method

### 3.1. Methodology Overview

SimGuard corrupts invoice features to generate contrastive pairs, trains a self-supervised encoder with an asymmetric, temperature-scaled InfoNCE loss, and follows a curriculum schedule that transitions from broad semantic clustering to fine-grained anomaly separation. Embedded invoices are then compared to their top-$k$ vendor-scoped neighbors to derive few-shot similarity-weighted risk scores. Table 2 summarizes the notation.

**Contrastive Task.** We adapt SCARF to tabular data. For each numerical feature $j$ of invoice $x_i$, we generate a positive pair $(x_i, \tilde{x}_i)$ by corrupting each feature independently with probability $p_{\text{corr}}$:

$$\tilde{x}_{i,j} = \begin{cases} \text{Uniform}(\min_j, \max_j), & \text{w.p. } p_{\text{corr}} \\ x_{i,j}, & \text{w.p. } 1 - p_{\text{corr}}, \end{cases} \quad (1)$$

where $\min_j, \max_j$ are the per-feature range, while categorical features are encoded as one-hot vectors and held out from corruption ($p_{\text{corr}}(x_j^{\text{cat}}) = 0$), serving as anchors. The per-feature alignment decomposition and dispersion-gradient analysis are in App. C.4; a worked feature-scale example is in App. E.4.

### 3.2. Asymmetric Temperature Scaling

As the number of negatives $m \to \infty$, SimGuard's asymmetric InfoNCE decomposes (up to constants) into an alignment term and a dispersion term:

$$\mathcal{L}_{\text{InfoNCE}} = \underbrace{\frac{1}{2\tau_{\text{num}}} \mathbb{E}_{\mathbf{x}, \tilde{\mathbf{x}}} \|\phi(\mathbf{x}) - \phi(\tilde{\mathbf{x}})\|^2}_{\text{Alignment}} \\ + \underbrace{\mathbb{E}_{\mathbf{x}} \log \mathbb{E}_{\mathbf{x}^-} e^{\phi(\mathbf{x})^\top \phi(\mathbf{x}^-)/\tau_{\text{denom}}}}_{\text{Dispersion}} \quad (2)$$

Theorem 4 proves this convergence (full proof in App. E.5). The decomposition isolates two levers: $\tau_{\text{num}}$ controls *alignment* weight (invariance to corruption), and $\tau_{\text{denom}}$ controls *dispersion* sharpening (hard-negative emphasis). SimGuard operates in the **asymmetric regime** $\tau_{\text{denom}} \ll \tau_{\text{num}}$: higher $\tau_{\text{num}}$ yields broad alignment (invariance to irrelevant perturbations), while lower $\tau_{\text{denom}}$ yields concentrated dispersion (sharp discrimination of subtle error patterns). Setting

*Table 1.* **SimGuard matches all specs:** feature comparison against representative baselines for large-scale tabular error detection. ✓ = property satisfied.

| Property | Random Forest | XGBoost | TabPFN | SCARF | SimGuard |
|---|---|---|---|---|---|
| Handles high-dim tabular data | ✓ | ✓ | ✓ | ✓ | ✓ |
| Self-supervised (no labeled anomalies needed) | | | | ✓ | ✓ |
| Context-aware (vendor-scoped retrieval) | | | | | ✓ |
| Few-shot adaptation without retraining | | | ✓ | | ✓ |
| Categorical-feature anchoring | | | | | ✓ |
| Asymmetric InfoNCE temperatures | | | | | ✓ |
| Case-level explainability via neighbors | | | | | ✓ |

*Table 2.* **Symbols and definitions.**

| Symbol | Definition |
|---|---|
| $x, \tilde{x}$ | input invoice; SCARF-corrupted variant |
| $\phi$ | encoder, $\phi : \mathcal{X} \to S^{d-1}$ |
| $\tau_{\text{num}}, \tau_{\text{denom}}$ | numerator / denominator temperatures |
| $p_{\text{corr}}$ | per-feature corruption probability |
| $\mathcal{N}_k(x_q)$ | $k$-NN of $x_q$ (vendor-scoped) |
| $k$ | neighbor count ($k$=15) |
| $\theta$ | similarity threshold ($\theta$=0.8) |
| $k_{\text{min}}$ | min evidence count (default 3) |
| $Y(x_j)$ | error label of case $x_j$ |
| $\widehat{R}_k$ | WeightedRisk (Eq. 3) |
| $p_{\text{err}}(x)$ | posterior $\mathbb{P}(Y{=}1 \mid x)$ |

$p_{\text{corr}}(x_j^{\text{cat}}) = 0$ makes categorical features act as anchors: the alignment penalty then couples only through numerical features, leaving categorical variation free to drive discrimination through the dispersion term (Theorem 6). As $\tau_{\text{denom}} \to 0$ the dispersion gradient concentrates on the hardest negative, amplifying subtle feature distinctions critical for error detection. The full per-feature alignment decomposition, dispersion-gradient analysis, and the temperature-effect visualization (Fig. 2 in App. C.4) are deferred to the appendix.

### 3.3. Few-Shot Risk Detection

For a new anomalous invoice $x_q$, SimGuard searches the historical repository of its vendor for similar past invoices. We compute its embedding $\phi(x_q)$ and retrieve its neighborhood $\mathcal{N}_q := \mathcal{N}(x_q) = \text{TopK}(x_q)$ of the $k$ most similar historical invoices, each with $\text{sim}(\phi(x_q), \phi(x_j)) > \theta$. Writing $w_j := \text{sim}(\phi(x_q), \phi(x_j))$ and $y_j := \mathbf{1}[\text{Error}(x_j)]$, the two risk scores are:

$$\text{BinaryRisk}(x_q) = \max_{x_j \in \mathcal{N}_q} y_j, \quad (3)$$

$$\text{WeightedRisk}(x_q) = \frac{\sum_{x_j \in \mathcal{N}_q} w_j \, y_j}{\sum_{x_j \in \mathcal{N}_q} w_j}. \quad (4)$$

Theorem 8 and Remark 9 establish that WeightedRisk is consistent and that BinaryRisk's false-negative probability decays exponentially in $k$. Retrieval is restricted within-

vendor; if $|\mathcal{N}(x_q)| < k_{\text{min}}$ or $\max_{x_j} w_j < \theta$, SimGuard flags $x_q$ as a candidate novel pattern. Here $k_{\text{min}}$ is the minimum-evidence threshold (default $k_{\text{min}} = 3$), and the investigated label is then added to $\mathcal{H}$ without model retraining.

### 3.4. Curriculum-based Scheduling

Fixed $(\tau_{\text{denom}}, p_{\text{corr}})$ underperforms curriculum learning by **12% to 18%** on missed errors (Table 3). We alternate adjustments of a single parameter at a time across 50-epoch phases, progressively decreasing both $\tau_{\text{denom}}$ (from 1.0 to 0.1) and $p_{\text{corr}}$ (from 0.5 to 0.2). After 200 epochs we also add a penalty for dissimilar pairs ($\text{sim}(\phi(x_i), \phi(x_j)) < 0.4$) to emphasize hard negatives. The full per-phase schedule is in App. C.3. Theorem 6 and Theorem 4 formalize why $\tau_{\text{denom}} \ll \tau_{\text{num}}$ (we use "$\ll$" to mean $\tau_{\text{denom}}/\tau_{\text{num}} \leq 0.1$ throughout) preserves rare-error recall.

We report three metrics: Volume Reduction (%), $\Delta$Recall decrease (%), and F1 (defined in App. C.2). Implementation details (architecture, optimizer, schedule) are in App. C.1.

## 4. Experimental Results

We evaluate SimGuard on two proprietary enterprise-scale invoice datasets (Model 1: 450k invoices; Model 2: 800k invoices) and three public fraud benchmarks (Santander (Piedra et al., 2019), Credit Card Fraud, IEEE Fraud (Howard et al., 2019)). Experimental setup and implementation details (architecture, optimizer, $k$=15, $\theta$=0.8, 3-day leakage gap) are in App. C.1; full ablation tables are in App. D.1. Experiments answer three questions, one per contribution:

- **Q1 (Asymmetry).** Does asymmetric InfoNCE with $\tau_{\text{denom}} \ll \tau_{\text{num}}$ outperform fixed-$\tau$ SCARF?

- **Q2 (Curriculum).** Does a 50-epoch alternating schedule over $(\tau_{\text{denom}}, p_{\text{corr}})$ beat fixed-parameter training?

- **Q3 (Adaptivity).** Does SimGuard's few-shot similarity retrieval generalize to public fraud benchmarks?

**Q1 & Q2 (Asymmetry, Curriculum).** Table 3 shows that

*Table 3.* **SimGuard wins (enterprise):** missed invoices (false negatives) at fixed volume reductions ($\Delta V$) under different temperature/corruption strategies, using WeightedRisk. Lower is better; bold = best. Full 12-row table in App. D.1, Table 6.

| Strategy ($\tau_{denom}, p_{corr}$) | Model 1 ($\Delta V$) | | | Model 2 ($\Delta V$) | | |
|---|---|---|---|---|---|---|
| | 10% | 20% | 30% | 10% | 20% | 30% |
| baseline (1.0, 0.5, no penalty) | 6 | 16 | 31 | 4 | 16 | 32 |
| fixed (1.0, 0.5) | 9 | 17 | 32 | 3 | 13 | 31 |
| $p_{corr} \downarrow 0.2$ only | 7 | 14 | 29 | 5 | 14 | 30 |
| $\tau_{denom} \uparrow 5, p_{corr} \downarrow$ | 12 | 17 | 31 | 3 | 13 | 31 |
| $\tau_{denom} \downarrow 0.1, p_{corr} \downarrow$ | **6** | **14** | **24** | **1** | **11** | **23** |
| $\quad + \tau_{num} = 2$ | **5** | **14** | 32 | **0** | **10** | **24** |

*Table 4.* **Public benchmarks:** performance across datasets with different class imbalance. Class 1 is the minority (error) class. SimGuard improves F1 over TabPFN on Santander and Credit Card Fraud.

| Model | Santander (10%) | | | Credit Card (0.17%) | | | IEEE Fraud (3.59%) | | |
|---|---|---|---|---|---|---|---|---|---|
| | P | R | F1 | P | R | F1 | P | R | F1 |
| Random Forest | 0.36 | 0.40 | 0.38 | 0.84 | 0.77 | **0.80** | 0.35 | **0.70** | **0.47** |
| XGBoost | 0.35 | 0.63 | 0.45 | 0.77 | 0.79 | 0.78 | 0.31 | **0.81** | 0.45 |
| TabPFN | 0.49 | 0.53 | 0.51 | 0.63 | **0.87** | 0.73 | 0.77 | 0.25 | 0.38 |
| **SimGuard** (Weighted) | 0.55 | 0.51 | **0.53** | 0.69 | **0.87** | 0.77 | 0.78 | 0.25 | 0.38 |
| SimGuard (KNN) | **0.65** | 0.34 | 0.45 | 0.80 | 0.76 | 0.78 | **0.80** | 0.24 | 0.37 |

decreasing $\tau_{denom}$ to 0.1 with a decreasing corruption schedule reduces missed errors from 31 or 32 down to **23 or 24** at 30% volume reduction on both enterprise datasets. The corruption-only ablation (App. D.1, Table 5) shows weak sensitivity to $p_{corr}$ alone, confirming that joint ($\tau_{denom}, p_{corr}$) scheduling is the driver; joint scheduling reduces $\Delta$Recall by **12% to 18%** versus fixed baselines (App. D.1, Table 6). Adding $\tau_{num} = 2$ (last row) further cuts Model 2's 10% volume missed errors to **zero** but degrades Model 1's 30% volume case, indicating that the optimal $\tau_{num}$ is dataset-dependent (full sweep in App. D.1).

**Q3 (Adaptivity).** Table 4 shows SimGuard generalizes: F1 of **0.53** on Santander (vs TabPFN 0.51), **0.77 to 0.78** on Credit Card Fraud with recall **0.87**, and competitive precision on IEEE Fraud. The Weighted variant matches TabPFN's recall while raising precision (0.63 to 0.69 on Credit Card), showing similarity-weighted voting filters false positives without dropping true errors. IEEE Fraud remains the hardest regime: tree ensembles retain a recall edge because bagged splits exploit raw tabular features when the fraud class is neither rare nor well-clustered. Overall, SimGuard acts as a false-positive-reduction layer on a high-recall first-stage classifier without per-dataset retraining.

## 5. Conclusion

We presented **SimGuard**, a similarity-guided error-detection framework for financial transactions that delivers three wins, matching the three contributions from the introduction:

- **Asymmetry.** Asymmetric InfoNCE with decoupled $\tau_{num}$ and $\tau_{denom}$ prevents feature suppression on heterogeneous tabular data (Theorem 4).

- **Curriculum.** A 50-epoch alternating schedule over ($\tau_{denom}, p_{corr}$) reduces missed errors by 12% to 18% and $\Delta$Recall below 1% at 30% volume reduction.

- **Adaptivity.** Vendor-scoped few-shot retrieval absorbs new error patterns without retraining and yields case-level explanations, with consistency proven in Theorem 8.

Empirically SimGuard achieves a **15% reduction** in flagged anomalies with $<$**1% decrease** in recall on enterprise data, F1 of **0.53** vs TabPFN 0.51 on Santander, and **0.77 to 0.78** on Credit Card Fraud. SimGuard's principle, asymmetric-temperature contrastive learning with few-shot retrieval, extends naturally to other heterogeneous-tabular domains (healthcare, manufacturing) whenever a historical repository of investigator-verified cases is available.

**Limitations.** SimGuard requires a populated historical repository; performance degrades on genuinely novel attack patterns where no similar precedent exists, and on regimes with neither extreme class imbalance nor clear clustering (e.g., IEEE Fraud). Full discussion and future work in App. G.

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

## Appendix Overview

This appendix is organized to mirror the paper's flow.

## A. Extended Motivation and Existing-Solution Limitations

Financial anomaly detection faces a fundamental challenge: distinguishing legitimate variations from actual errors. The error landscape continuously evolves as perpetrators adapt their modus operandi (MOs), and anomaly models are constrained by their knowledge of varying contexts: vendor behavior varies across countries, regions, business types and industries. A seasonal billing spike may signal suspicious activity for a steady-service vendor but reflect expected patterns for a retailer during the holidays. Three limitations drive our design:

- **MO specialization vs. generalization.** Developing a specialized model for each evolving error pattern consumes excessive time and resources; models risk going stale as new MOs emerge.

- **Limited use of investigation history.** Existing unsupervised systems cannot leverage historical case resolutions, hindering continuous improvement from human feedback.

- **Representation challenges.** Financial data is heterogeneous tabular data (numerical, categorical, textual) with no natural similarity metric, complicating identification of semantically similar cases across different feature distributions and vendor behaviors.

Though our datasets are tabular, a theoretical insight underlying our approach recognizes that normal data typically lie on a low-dimensional manifold despite appearing as high-dimensional mixed-type features (Goyal et al., 2020). Non-embedding distance-based methods like $k$NN fail in high-dimensional spaces due to the curse of dimensionality, where points become equidistant.

## B. Extended Related Work

**Anomaly Detection.** Anomaly detection methods have evolved from classical statistical approaches to deep learning. Deep autoencoders (Zhou & Paffenroth, 2017) identify anomalies through reconstruction error, while one-class methods like DeepSVDD (Ruff et al., 2018) learn compact representations of normal data. Goyal et al. (2020) propose that normal data resides on a low-dimensional manifold, generating synthetic anomalies near manifold boundaries. Contrastive learning (Tack et al., 2020) and meta-learning (Jeong & Kim, 2020) typically require labeled anomalies or representative examples from base classes, constraints impractical in financial settings where error patterns continuously evolve.

**Deep Learning for Tabular Data.** Despite the prevalence of tabular data in critical domains, deep learning has historically underperformed gradient-boosted trees (Grinsztajn et al., 2022). Specialized architectures including TabNet (Arik & Pfister, 2021) and FT-Transformer (Gorishniy et al., 2021) attempt to bridge this gap; effective representation learning for mixed categorical and numerical features remains challenging without domain expertise (Borisov et al., 2022). We use TabPFN v2 (Hollmann et al., 2025) as an out-of-the-box classifier baseline on public datasets.

**Self-Supervised and Contrastive Learning.** Contrastive methods evolved from metric-learning approaches such as triplet loss (Hoffer & Ailon, 2015) and soft-nearest-neighbor loss (Hinton et al., 2012), which introduced temperature-scaled similarity. The InfoNCE loss from SimCLR (Chen et al., 2020) and CPC (van den Oord et al., 2018) formalized the approach; SCARF (Bahri et al., 2022) adapts it to tabular data with random feature corruption, keeping $\tau_{\text{denom}} = \tau_{\text{num}}$. Temperature-scaling dynamics (Robinson et al., 2021; Kukleva et al., 2023) highlight how $\tau$ modulates hard-negative emphasis; Dinu et al. (2025) demonstrate the effectiveness of decoupled temperatures for text representation learning.

**Comparison to Temperature Scheduling.** Unlike vision/text approaches (Kukleva et al., 2023), SimGuard's joint temperature/corruption curriculum for tabular data synergistically modulates $\tau_{\text{denom}}$ and $p_{\text{corr}}$ to prevent financial feature suppression, and uses a decreasing schedule ($\tau_{\text{denom}} \downarrow$, $p_{\text{corr}} \downarrow$) to shift from broad feature patterns to fine-grained anomaly detection. This co-scheduling reduces false negatives by 12% to 18% vs. fixed-corruption baselines (Appendix D.1).

# C. Method Details

This section groups the implementation, metric definitions, curriculum schedule, and temperature-effect analyses deferred from §3.

## C.1. Implementation Details

SimGuard's encoder uses a 4-layer MLP for numerical features and a 2-layer network for categorical features before concatenation, both with BatchNorm and Dropout. We train with Adam (lr = 0.001) for 480 epochs (batch size 1024) on the proprietary datasets, and 150 epochs on the smaller public datasets. After 200 epochs we activate a penalty for dissimilar pairs (cosine similarity $< 0.4$) to emphasize hard negatives. For enterprise data, to prevent leakage the historical repository only includes transactions $\geq 3$ days before evaluation. Neighbor count $k = 15$ and similarity threshold $\theta = 0.8$ reflect business risk tolerance from preliminary experiments.

## C.2. Metric Definitions

We evaluate SimGuard using three complementary metrics:

- **Volume Reduction (%).** Decrease in flagged transactions requiring investigation: $\frac{|\mathcal{A}| - |\mathcal{A}_{\text{filtered}}|}{|\mathcal{A}|} \cdot 100\%$, where $\mathcal{A}$ is the original set of anomalies and $\mathcal{A}_{\text{filtered}}$ is the reduced set after SimGuard's filtering.

- **$\Delta$Recall Decrease (%).** Proportion of true errors missed under SimGuard's filtering vs. unfiltered baseline; lower is better.

- **F1 Score.** Harmonic mean of precision and recall on public benchmarks.

Varying the risk threshold on WeightedRisk traces a precision/recall curve across operational points.

## C.3. Curriculum Schedule

For the proprietary datasets, SimGuard follows the full 50-epoch alternating schedule: epochs 1 to 200 at ($\tau_{\text{denom}}{=}1.0, p_{\text{corr}}{=}0.5$) for stabilization, then $\tau_{\text{denom}} \rightarrow 0.8$ (201 to 250), $p_{\text{corr}} \rightarrow 0.4$ (251 to 300), $\tau_{\text{denom}} \rightarrow 0.5$ (301 to 350), $p_{\text{corr}} \rightarrow 0.3$ (351 to 400), $\tau_{\text{denom}} \rightarrow 0.1$ (401 to 450), $p_{\text{corr}} \rightarrow 0.2$ (451 to 480). We deliberately adjust only one parameter at a time to promote training stability; the 50-epoch phases provide adaptation time, creating smoother optimization trajectories for heterogeneous tabular data.

## C.4. Temperature Effects on Embeddings

**Extended alignment analysis.** The alignment term decomposes per feature as

$$\mathbb{E}_{\mathbf{x},\tilde{\mathbf{x}}}\|\phi(\mathbf{x}) - \phi(\tilde{\mathbf{x}})\|^2 = p_{\text{corr}} \sum_{j=1}^{n} \mathbb{E}_{\mathbf{x}}\|\phi(\mathbf{x}) - \phi(\tilde{\mathbf{x}}_j)\|^2,$$

where $\tilde{\mathbf{x}}_j$ is $\mathbf{x}$ with only feature $j$ corrupted. Lower $p_{\text{corr}}$ reduces the weighting of corrupted features, making the encoder more responsive to changes in those features.

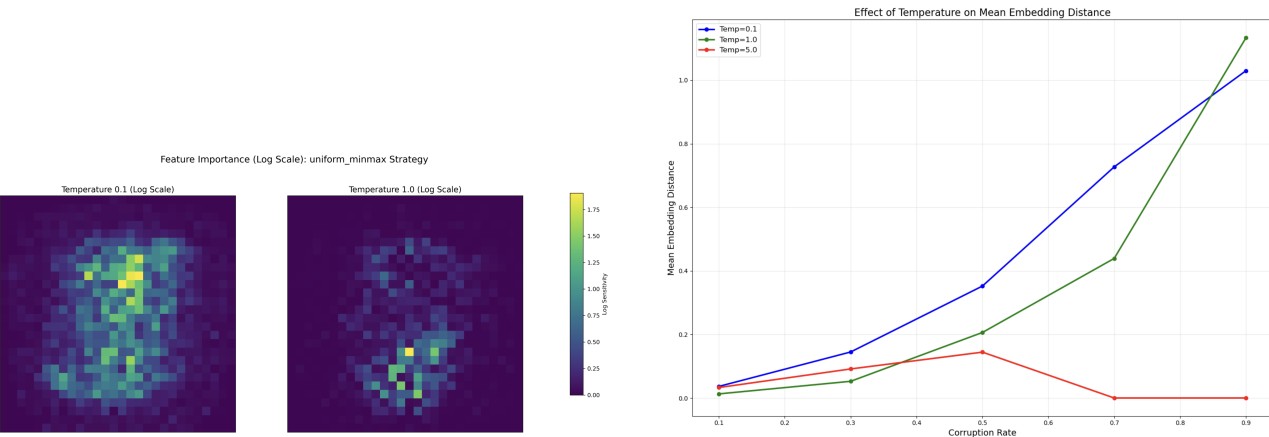

*(a)* **Asymmetry matters:** MNIST feature-importance map. Low $\tau_{denom} = 0.1$ (left) focuses on digit nuances; $\tau_{denom} = 1$ (right) has diffuse attention.

*(b)* **Sharper dispersion:** $\tau_{denom} \in \{0.1, 1, 5\}$ (blue, green, red). Lower $\tau_{denom}$ increases repulsion between dissimilar samples across corruption rates.

*Figure 2.* **Temperature effects:** lower $\tau_{denom}$ makes the model more sensitive to subtle feature differences and increases embedding separation between original/augmented samples.

**Categorical anchoring.** Setting $p_{\text{corr}}(x_j^{\text{cat}}) = 0$ for categorical features yields

$$\mathbb{E}_{\mathbf{x},\tilde{\mathbf{x}}}\|\phi(\mathbf{x}) - \phi(\tilde{\mathbf{x}})\|^2 = p_{\text{corr}} \sum_{j \in \mathcal{N}_{\text{num}}} \mathbb{E}_{\mathbf{x}}\|\phi(\mathbf{x}) - \phi(\tilde{\mathbf{x}}_j)\|^2.$$

Theorem 6 shows this first-order decomposition leaves the encoder free to distinguish categorical values through the dispersion term.

**Dispersion gradient.** For $\phi(\mathbf{x})^\top \phi(\mathbf{x}^-) \gg \tau_{\text{denom}}$,

$$\nabla_\phi \mathcal{L}_{\text{disp}} \propto \frac{1}{\tau_{\text{denom}}} \mathbb{E}_{\mathbf{x},\mathbf{x}^-}\left[\phi(\mathbf{x}^-) - \phi(\mathbf{x})\right],$$

so lower $\tau_{\text{denom}}$ amplifies the contribution of hard negatives. Fig. 2a visualizes this at $\tau_{denom} = 0.1$ vs. $1.0$ on MNIST, and Fig. 2b confirms it quantitatively across corruption rates.

# D. Additional Experiments

## D.1. Ablation Tables

# E. Theoretical Foundations

This section gives the full theoretical development: an informal summary of the Robinson (2021) feature-suppression phenomenon (§E.1), the formal generative model and propositions it rests on (§E.2), intuition for our asymmetric-temperature setup (§E.3), a worked numerical example (§E.4), and the three formal theorems with complete proofs that underlie SimGuard (§E.5).

## E.1. Robinson's Feature-Suppression Summary

This subsection reviews the key theoretical insights from Robinson (2021), which demonstrate how the standard InfoNCE objective can admit "shortcut" encodings that ignore some latent features. We summarize the definitions of feature suppression versus distinction, outline the infinite-negatives analysis of the InfoNCE loss and its two main propositions exposing this failure mode, and describe the Implicit Feature Modification (IFM) technique designed to force the encoder to recover all informative factors (Robinson et al., 2021).

1. **Feature Suppression under InfoNCE.** Definition 1 and Proposition 1 show that, even with infinitely many negatives,

*Table 5.* **Corruption alone is weak:** ablation runs isolating the effect of corruption schedules at fixed volume reductions, with $\tau_{denom}$ held constant at 1.0. Number of missed invoices (false negatives) at three volume-reduction thresholds (10%, 20%, 30%) using WeightedRisk.

|  | Missed invoices | | |
| --- | --- | --- | --- |
| Corruption strategy ($p_{corr}$) | 10% | 20% | 30% |
| *Model 1* | | | |
| $p_{corr} = 0.8$ | 4 | 16 | 29 |
| $p_{corr} = 0.5$ | 9 | 17 | 32 |
| $p_{corr} = 0.2$ | 6 | 14 | 27 |
| $p_{corr}$ decreased to 0.2 | 7 | 14 | 29 |
| $p_{corr}$ increased to 0.8 | 5 | 15 | 29 |
| *Model 2* | | | |
| $p_{corr} = 0.8$ | 6 | 14 | 28 |
| $p_{corr} = 0.5$ | 2 | 12 | 29 |
| $p_{corr} = 0.2$ | 2 | 15 | 33 |
| $p_{corr}$ decreased to 0.2 | 5 | 14 | 30 |
| $p_{corr}$ increased to 0.8 | 4 | 14 | 27 |

*Table 6.* **Enterprise gains:** comparison of temperature schedules on downstream task performance using BinaryRisk. We report $\Delta$Recall decrease (%), missed erroneous invoices (false negatives), and volume-reduction percentage. Similarity threshold $\theta = 0.8$, top-$k = 15$. Bold rows indicate the best strategy for each model.

| Temperature strategy | $\Delta$Recall (%) | FN | Vol. red. (%) |
| --- | --- | --- | --- |
| *Model 1* | | | |
| $\tau_{\text{denom}} = 1.0$, $p_{corr} = 0.5$, no penalty (baseline) | 2.13 | 13 | 17.42 |
| $\tau_{\text{denom}} = 1.0$, $p_{corr} = 0.5$ | 2.29 | 14 | 17.01 |
| $\tau_{\text{denom}} = 1.0$, $p_{corr} \downarrow 0.2$ | 1.96 | 12 | 15.26 |
| $\tau_{\text{denom}} \uparrow 5$, $p_{corr} \downarrow 0.2$ | 3.12 | 19 | 21.08 |
| $\tau_{\text{denom}} \downarrow 0.1$, $p_{corr} \downarrow 0.2$ | **0.81** | **5** | 9.26 |
| *Model 2* | | | |
| $\tau_{\text{denom}} = 1$, $p_{corr} = 0.5$, no penalty (baseline) | 1.50 | 8 | 12.81 |
| $\tau_{\text{denom}} = 1$, $p_{corr} = 0.5$ | 0.94 | 5 | 11.81 |
| $\tau_{\text{denom}} = 1.0$, $p_{corr} \downarrow 0.2$ | 1.12 | 6 | 13.43 |
| $\tau_{\text{denom}} \uparrow 5$, $p_{corr} \downarrow 0.2$ | 0.94 | 5 | 11.80 |
| $\tau_{\text{denom}} \downarrow 0.1$, $p_{corr} \downarrow 0.2$ | **0.56** | **3** | 11.48 |

optimal solutions exist that either suppress or distinguish any given feature, so minimizing InfoNCE alone does not guarantee retention of all predictive attributes.

2. **Control via Instance Discrimination Difficulty.** Proposition 2 (a restated, formal version of Robinson's Proposition 2) shows that when a feature subset $S$ is held constant across both positives and negatives in a batch, any minimizer of the conditional InfoNCE loss must suppress $S$. Lowering the temperature $\tau$ (or sampling harder negatives) shifts gradient emphasis onto difficult pairs, recovering suppressed features at the cost of potentially suppressing others: a fundamental trade-off.

3. **Implicit Feature Modification (IFM).** To address this, IFM applies small adversarial perturbations in embedding space to remove the most-used features, compelling the encoder to capture additional factors without discarding the originals.

### E.2. Generative Model and Propositions

**Data Generative Model.** Let each invoice be generated by $n$ independent latent factors

$$z = (z_1, \ldots, z_n) \ \in \ Z = \prod_{j=1}^{n} Z_j,$$

where, for example, $z_1$ is the invoice amount, $z_2$ the invoice date, $z_3$ the vendor category, and so on, each distributed according to density $p_j$. The observed invoice record $x \in X$ is produced by an injective map

$$g : Z \to X, \quad x = g(z),$$

so that in principle one could recover all latent factors from $x$.

**Contrastive Encoder.** *Notation.* In this subsection we follow Robinson et al.'s notation (encoder $f$, embedding $v$) for fidelity to their original propositions; elsewhere in the paper we write $\phi$ for the encoder. A self-supervised encoder

$$f : X \to S^{d-1} \subset \mathbb{R}^d$$

projects each invoice $x$ to a unit-norm embedding $v = f(x)$. We denote by

$$v^+ = f(x^+), \quad v_i^- = f(x_i^-)$$

the embeddings of a "positive" variant $x^+$ (e.g. with a corrupted amount) and "negative" examples $x_i^-$.

**Feature Subsets and Embedding Regions.** To test whether the encoder uses particular invoice attributes, fix an index set $S \subset \{1, \dots, n\}$ (e.g. $S = \{1\}$ for amount, or $S = \{2, 3\}$ for date and vendor). Write

$$z_S = (z_j)_{j \in S}, \quad z_{\bar{S}} = (z_j)_{j \notin S},$$

so $z = (z_S, z_{\bar{S}})$. Let $\lambda$ be the joint density on $Z$. For any measurable region

$$E \subset S^{d-1},$$

such as a small spherical cap around the embedding of a typical invoice, define

$$\mu\big(E \mid z_S\big) = \lambda\big\{ z_{\bar{S}} \in Z_{\bar{S}} : f\big(g(z_S, z_{\bar{S}})\big) \in E\big\},$$

the conditional probability that the embedding falls in $E$ given fixed values of the attributes in $S$.

1. $f$ *suppresses* $S$ if
$$\mu(\cdot \mid z_S) = \mu(\cdot \mid \bar{z}_S) \quad \text{for all } z_S, \bar{z}_S \in Z_S,$$

   meaning that changing those invoice attributes does not alter the embedding distribution. *Example:* If $S = \{\text{amount}\}$, suppression implies that small and large amounts yield the same embedding patterns.

2. $f$ *distinguishes* $S$ if, whenever $z_S \neq \bar{z}_S$, the supports of $\mu(\cdot \mid z_S)$ and $\mu(\cdot \mid \bar{z}_S)$ are disjoint, so different values of the attributes in $S$ map to non-overlapping embedding regions. *Example:* If $S = \{\text{vendor category}\}$, distinguishing means invoices from different vendor types cluster separately.

**InfoNCE Objective.** Given a batch $(x, x^+, \{x_i^-\}_{i=1}^m)$, define

$$\ell\big(v, v^+, \{v_i^-\}\big) = -\log \frac{\exp\big(v^\top v^+/\tau\big)}{\exp\big(v^\top v^+/\tau\big) + \sum_{i=1}^m \exp\big(v^\top v_i^-/\tau\big)}.$$

As $m \to \infty$, this converges (up to constants) to

$$L(f) = \frac{1}{2\tau}\, \mathbb{E}_{x,x^+} \|f(x) - f(x^+)\|^2 + \mathbb{E}_{x^+}\Big[\log \mathbb{E}_{x^-} \exp\big(f(x^+)^\top f(x^-)/\tau\big)\Big].$$

**Main Propositions.**

**Proposition 1** (Robinson 2021, Prop. 1). *When each latent $z_j$ (e.g. amount, date, vendor) is uniform on its domain and independent, there exist two minimizers of $L(f)$: one that suppresses a selected attribute subset $S \subseteq [n]$ and another that distinguishes it. Both attain the same infimum of the limiting InfoNCE loss.*

*Proof idea.* Under independent uniform latents the InfoNCE infimum is attained by any encoder for which the induced pushforward on $S^{d-1}$ is uniform. Two such encoders exist: one that folds $S$-variation into the fibers (suppresses $S$) and one that spreads $S$-variation into disjoint embedding regions (distinguishes $S$). Both produce the same uniform pushforward, hence the same loss. The full proof is in Robinson et al. (2021, App. A). □

**Proposition 2** (Robinson 2021, Prop. 2, restated). *Fix $S \subseteq [n]$ and let each marginal $p_j$ be uniform on $Z_j = S^{d-1}$. Suppose the positive pair $(x, x^+)$ and every negative $x_i^-$ in a batch are conditioned to share the same value of $z_S$ (i.e., $z_S$ is held constant across the entire batch). Then any $f$ that minimizes the corresponding limiting conditional InfoNCE loss suppresses $S$, in the sense that $\mu(\cdot \mid z_S) = \mu(\cdot \mid \bar{z}_S)$ almost surely.*

*Proof sketch.* Under the stated conditioning, every embedding in the batch, whether positive or negative, is a deterministic function of $z_{\bar{S}}$ once $z_S$ is fixed. The dispersion term $\mathbb{E}_{x^+} \log \mathbb{E}_{x^-} \exp(f(x^+)^\top f(x^-)/\tau)$ therefore becomes invariant to the value of $z_S$ in the batch, and so it cannot generate gradient signal that separates distinct $z_S$-values. The alignment term $(2\tau)^{-1}\mathbb{E}\|f(x) - f(x^+)\|^2$ attains its infimum at any encoder with $f(x) = f(x^+)$ on positive pairs, and this is achievable by an encoder that is constant in $z_S$. Thus a $z_S$-invariant encoder attains the infimum, and by the rigidity argument of Robinson et al. (2021, App. A) any other minimizer must agree with it on the $z_S$-conditioned pushforwards, which is suppression in the sense of Definition 1. □

*Remark* 3. Proposition 2 is the formal statement of the result loosely summarized as "Proposition 2 (Informal)" in earlier versions of this manuscript. The conditioning is on both *positives and negatives* sharing $z_S$; if only positives share $z_S$ while negatives are drawn i.i.d. from $p_{\text{data}}$, the dispersion term does produce a separating signal and the conclusion does *not* apply. This distinction is central to Theorem 6: our categorical-anchored SCARF setup holds categorical values constant across the positive pair but samples negatives i.i.d., escaping the hypothesis of Proposition 2.

**Key Takeaway.** These results explain why, without mechanisms like asymmetric temperature scaling, a contrastive encoder may ignore important invoice attributes, favoring "shortcut" embeddings that hamper anomaly detection.

### E.3. Contrastive Learning with Feature Corruption (Intuition)

This subsection presents the intuition behind asymmetric temperature decoupling. The formal results are stated and proved in §E.5.

#### E.3.1. MODEL SETUP

Let $x \in \mathcal{X}$ be an invoice with numerical features $x_j$ ($j = 1, \ldots, n$). The encoder $\phi : \mathcal{X} \to S^{d-1}$ maps $x$ to a unit-norm embedding. We generate corrupted positives $\tilde{x}$ as:

$$\tilde{x}_j = \begin{cases} \text{Uniform}(\min_j, \max_j) & \text{with probability } p_{\text{corr}}, \\ x_j & \text{with probability } 1 - p_{\text{corr}}, \end{cases}$$

with corruption rate $p_{\text{corr}}$. Let $\tau_{\text{num}}, \tau_{\text{denom}}$ be the numerator and denominator temperatures.

#### E.3.2. ASYMPTOTIC INFONCE LOSS WITH CORRUPTION

For $m \to \infty$ negatives, the loss converges to:

$$L(\phi) = \underbrace{\frac{1}{2\tau_{\text{num}}}\mathbb{E}_{x,\tilde{x}} \left[\|\phi(x) - \phi(\tilde{x})\|^2\right]}_{\text{Alignment}} + \underbrace{\mathbb{E}_x \log \mathbb{E}_{x^-} \exp\left(\frac{\phi(x)^\top \phi(x^-)}{\tau_{\text{denom}}}\right)}_{\text{Dispersion}}.$$

**Alignment Term Analysis.** The alignment term enforces invariance to corruptions:

$$\mathbb{E}_{x,\tilde{x}}\|\phi(x) - \phi(\tilde{x})\|^2 = p_{\text{corr}} \sum_{j=1}^n \mathbb{E}_x\|\phi(x) - \phi(\tilde{x}_j)\|^2,$$

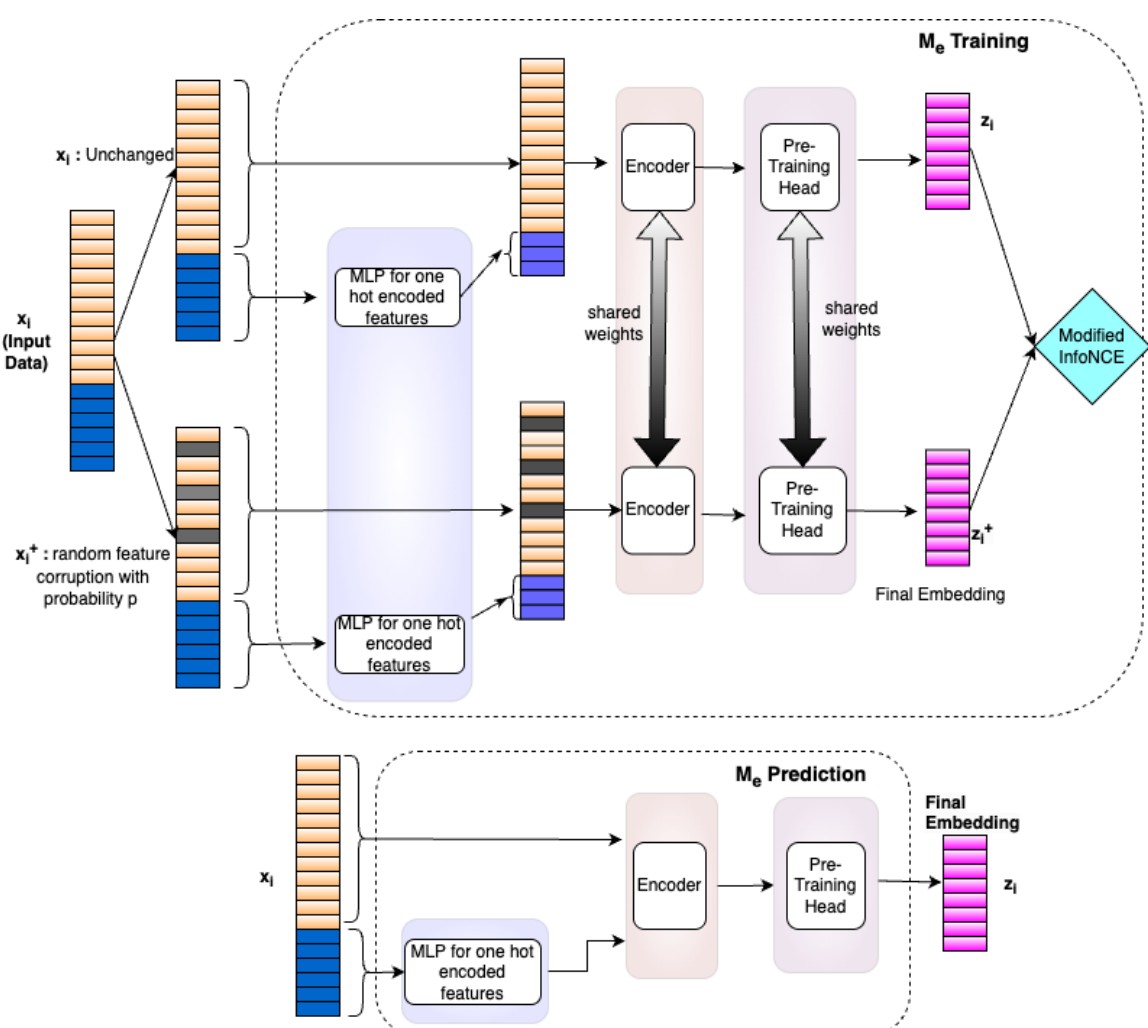

*Figure 3.* **Backbone architecture:** SCARF encoder used by SimGuard for tabular data.

where $\tilde{x}_j$ is $x$ with only feature $j$ corrupted. Lower $p_{\text{corr}}$ reduces the weighting of corrupted features in the loss, making the encoder *less* invariant to those features.

**Dispersion Term Analysis.** The dispersion term penalizes embedding overlap with negatives. For $\tau_{\text{denom}} \to 0$, $\exp(\phi(x)^\top \phi(x^-)/\tau_{\text{denom}})$ approaches a Dirac delta at the hardest negative, sharpening cluster boundaries.

### E.3.3. HYPERPARAMETER TRADEOFFS

**Case: Decreasing $p_{\text{corr}}$ (0.5 → 0.2) & Decreasing $\tau_{\text{denom}}$**

- **Lower $p_{\text{corr}}$**: Reduces the alignment term's focus on corrupted features. The encoder preserves more information about uncorrupted features (e.g., invoice amount or vendor), improving *distinction* for attributes rarely perturbed.

- **Lower $\tau_{\text{denom}}$**: Amplifies the dispersion term's penalty for overlapping negatives. Embeddings of dissimilar invoices (e.g., different vendors) are pushed farther apart, enhancing cluster separation.

**Example**: Suppose $S = \{\text{vendor}\}$ is a key attribute. Lowering $p_{\text{corr}}$ preserves vendor information in embeddings, while reducing $\tau_{\text{denom}}$ ensures invoices from different vendors cluster separately. This combats "over-suppression" of $S$ when $p_{\text{corr}}$ is too high.

### E.4. Worked Example: Asymmetric Feature Normalization

This worked example concretizes Theorem 6 with specific feature ranges, showing how a narrow-range feature contributes only $\sim 23\%$ of the alignment loss compared to a full-range feature.

### E.4.1. PROBLEM SETUP

Consider a dataset with features normalized as follows:

- **Narrow Feature (j=1)**: $x_1 \in [a, 1]$, where $a = 0.52$.

- **Full-Range Features** ($j \geq 2$): $x_j \in [0, 1]$.

Corruption is applied uniformly with rate $p_{\text{corr}}$:

$$\tilde{x}_j = \begin{cases} \text{Uniform}(a, 1) & \text{for } j = 1, \text{ w.p. } p_{\text{corr}} \\ \text{Uniform}(0, 1) & \text{for } j \geq 2, \text{ w.p. } p_{\text{corr}} \\ x_j & \text{otherwise.} \end{cases}$$

### E.4.2. ALIGNMENT LOSS DECOMPOSITION

The alignment term becomes:

$$L_{\text{align}} = \frac{p_{\text{corr}}}{2\tau_{\text{num}}} \left( \underbrace{\mathbb{E}_{x_1} \|\phi(x) - \phi(\tilde{x}_1)\|^2}_{\text{Narrow Feature Term}} + \sum_{j=2}^{n} \underbrace{\mathbb{E}_{x_j} \|\phi(x) - \phi(\tilde{x}_j)\|^2}_{\text{Full-Range Feature Term}} \right).$$

**Narrow Feature Term Analysis.** Let $\Delta_1 = 1 - a = 0.48$. The maximum input perturbation is:

$$\max |\tilde{x}_1 - x_1| \leq \Delta_1.$$

Assuming Lipschitz continuity of $\phi$ with constant $K$:

$$\mathbb{E}\|\phi(x) - \phi(\tilde{x}_1)\|^2 \leq K^2 \mathbb{E}|\tilde{x}_1 - x_1|^2 = K^2 \cdot \frac{\Delta_1^2}{3}. \quad \text{(Var of Uniform: } \frac{\Delta^2}{12})$$

Thus:

$$\text{Narrow Feature Term} \propto \frac{\Delta_1^2}{3} \approx 0.0768 K^2.$$

**Full-Range Feature Term Analysis.** For $j \geq 2$, $\Delta_j = 1 - 0 = 1$:

$$\mathbb{E}\|\phi(x) - \phi(\tilde{x}_j)\|^2 \leq K^2 \cdot \frac{\Delta_j^2}{3} = \frac{K^2}{3} \approx 0.333K^2.$$

### E.4.3. ALIGNMENT LOSS RATIO

The narrow feature contributes:

$$\frac{L_{\text{align},1}}{L_{\text{align},j\geq 2}} \approx \frac{0.0768}{0.333} \approx 0.23.$$

**Interpretation**: The narrow feature contributes 23% of the alignment loss per full-range feature, leading to weaker suppression. This numerical example is consistent with the general Lipschitz bound $\mathbb{E}\|\phi(x) - \phi(\tilde{x}_j)\|^2 \leq K^2\Delta_j^2/3$, which shows that per-feature alignment penalty scales quadratically with the feature range $\Delta_j$; the $p_{\text{corr}}$-weighted, categorical-anchored sum of these per-feature terms is the right-hand side of Theorem 6, Eq. (9).

### E.4.4. DISPERSION TERM IMPACT

The dispersion term:

$$L_{\text{disp}} = \mathbb{E}_x \log \mathbb{E}_{x^-} \exp\left(\frac{\phi(x)^\top \phi(x^-)}{\tau_{\text{denom}}}\right),$$

is dominated by feature $j$ with greater variability. For unit-norm embeddings:

$$\phi(x)^\top \phi(x^-) \approx 1 - \frac{1}{2}\|\phi(x) - \phi(x^-)\|^2.$$

Full-range features induce larger $\|\phi(x) - \phi(x^-)\|^2$, contributing more to dispersion.

### E.4.5. BALANCING MECHANISMS

**1. Per-Feature Normalization.** Normalize all features to $[0, 1]$:

$$\tilde{x}_j^{(\text{norm})} = \frac{x_j - \min_j}{\max_j - \min_j} \quad \forall j.$$

Ensures $\Delta_j = 1$ for all $j$, equalizing alignment contributions.

**2. Feature-Specific Temperature Scaling.** Assign lower $\tau_{\text{denom},1}$ to sharpen dispersion for the narrow feature:

$$\tau_{\text{denom},j} = \begin{cases} 0.1 & j = 1 \\ 0.5 & j \geq 2. \end{cases}$$

## E.5. Formal Theorems and Proofs

This section formalizes three claims that were stated informally in Sections 3 and in the earlier parts of this appendix. Throughout, we use the notation already established: $\phi : \mathcal{X} \to S^{d-1}$ is a measurable encoder onto the unit sphere, positive pairs $(x, \tilde{x}) \sim p_{\text{pos}}$ are produced by the SCARF corruption operator with per-feature corruption probability $p_{\text{corr}}(j)$, negatives $\{x_i^-\}_{i=1}^m$ are i.i.d. $\sim p_{\text{data}}$, and $\tau_{\text{num}}, \tau_{\text{denom}} > 0$ are the numerator and denominator temperatures. The batch asymmetric InfoNCE loss is

$$L_m(\phi; \tau_{\text{num}}, \tau_{\text{denom}}) = \mathbb{E}\left[-\log \frac{e^{\phi(x)^\top \phi(\tilde{x})/\tau_{\text{num}}}}{e^{\phi(x)^\top \phi(\tilde{x})/\tau_{\text{num}}} + \sum_{i=1}^m e^{\phi(x)^\top \phi(x_i^-)/\tau_{\text{denom}}}}\right], \tag{5}$$

where the expectation is over $(x, \tilde{x}) \sim p_{\text{pos}}$ and $\{x_i^-\}_{i=1}^m$ i.i.d. $\sim p_{\text{data}}$, independent of the positive pair.

E.5.1. ASYMPTOTIC DECOMPOSITION OF THE ASYMMETRIC INFONCE LOSS

**Theorem 4** (Asymmetric Alignment/Dispersion Decomposition). *Fix $\tau_{num}, \tau_{denom} > 0$, assume $\phi(x) \in S^{d-1}$ almost surely, and assume $\mathbb{E}_{x^-}\left[e^{\phi(x)^\top \phi(x^-)/\tau_{denom}}\right]$ is bounded and bounded away from $0$ uniformly in $x$ (which holds because $|\phi(x)^\top \phi(x^-)| \leq 1$). Then, as the number of negatives $m \to \infty$,*

$$
L_m(\phi; \tau_{num}, \tau_{denom}) - \log m - \frac{1}{\tau_{num}} \longrightarrow
$$

$$
\underbrace{\frac{1}{2\tau_{num}} \mathbb{E}_{(x,\tilde{x})\sim p_{pos}}\left[\|\phi(x) - \phi(\tilde{x})\|^2\right]}_{\text{Alignment } \mathcal{A}(\phi)}
$$

$$
+ \underbrace{\mathbb{E}_x\left[\log \mathbb{E}_{x^-\sim p_{data}} e^{\phi(x)^\top \phi(x^-)/\tau_{denom}}\right]}_{\text{Dispersion } \mathcal{D}(\phi)}.
$$

(6)

*Proof.* Write $a = \phi(x)^\top \phi(\tilde{x})$ and $b_i = \phi(x)^\top \phi(x_i^-)$. Expanding the logarithm,

$$
-\log \frac{e^{a/\tau_{\text{num}}}}{e^{a/\tau_{\text{num}}} + \sum_{i=1}^m e^{b_i/\tau_{\text{denom}}}} = -\frac{a}{\tau_{\text{num}}} + \log\left(e^{a/\tau_{\text{num}}} + \sum_{i=1}^m e^{b_i/\tau_{\text{denom}}}\right).
$$

Factoring $m$ inside the logarithm,

$$
\log\left(e^{a/\tau_{\text{num}}} + \sum_{i=1}^m e^{b_i/\tau_{\text{denom}}}\right) = \log m + \log\left(\frac{e^{a/\tau_{\text{num}}}}{m} + \frac{1}{m}\sum_{i=1}^m e^{b_i/\tau_{\text{denom}}}\right).
$$

Since $|b_i| \leq 1$ and $\tau_{\text{denom}} > 0$, the summands $e^{b_i/\tau_{\text{denom}}}$ are bounded in $[e^{-1/\tau_{\text{denom}}}, e^{1/\tau_{\text{denom}}}]$ and i.i.d. given $x$, so by the strong law of large numbers,

$$
\frac{1}{m}\sum_{i=1}^m e^{b_i/\tau_{\text{denom}}} \xrightarrow{\text{a.s.}} \mathbb{E}_{x^-\sim p_{\text{data}}}\left[e^{\phi(x)^\top \phi(x^-)/\tau_{\text{denom}}}\right] =: \psi(x).
$$

Simultaneously, $e^{a/\tau_{\text{num}}}/m \to 0$. Combining, the interior of the logarithm converges almost surely to $\psi(x)$, which is bounded and strictly positive by assumption; the continuous mapping theorem yields

$$
\log\left(\frac{e^{a/\tau_{\text{num}}}}{m} + \frac{1}{m}\sum_{i=1}^m e^{b_i/\tau_{\text{denom}}}\right) \xrightarrow{\text{a.s.}} \log \psi(x).
$$

Dominated convergence (whose hypothesis is met because the integrand is uniformly bounded in absolute value by $|\log \psi(x)| + O(1) \leq 1/\tau_{\text{denom}} + O(1)$) gives

$$
\mathbb{E}\left[\log\left(e^{a/\tau_{\text{num}}} + \sum_{i=1}^m e^{b_i/\tau_{\text{denom}}}\right)\right] - \log m \longrightarrow \mathbb{E}_x\left[\log \mathbb{E}_{x^-} e^{\phi(x)^\top \phi(x^-)/\tau_{\text{denom}}}\right] = \mathcal{D}(\phi).
$$

(7)

For the numerator term, the unit-sphere identity $\phi(x)^\top \phi(y) = 1 - \frac{1}{2}\|\phi(x) - \phi(y)\|^2$ applied to the positive pair gives

$$
-\mathbb{E}\left[\frac{a}{\tau_{\text{num}}}\right] = -\frac{1}{\tau_{\text{num}}}\mathbb{E}_{(x,\tilde{x})\sim p_{\text{pos}}}\left[\phi(x)^\top \phi(\tilde{x})\right] = -\frac{1}{\tau_{\text{num}}} + \frac{1}{2\tau_{\text{num}}}\mathbb{E}\left[\|\phi(x) - \phi(\tilde{x})\|^2\right] = -\frac{1}{\tau_{\text{num}}} + \mathcal{A}(\phi).
$$

(8)

Summing (7) and (8) and moving the $-\log m - 1/\tau_{\text{num}}$ constants to the left-hand side gives (6). $\square$

*Remark* 5. The symmetric case $\tau_{\text{num}} = \tau_{\text{denom}} = \tau$ reduces Theorem 4 to the limit of Wang and Isola (Wang & Isola, 2020) (their Theorem 1), and of Robinson et al. (Robinson et al., 2021) (Section 2.2 limit). The asymmetric extension shows that $\tau_{\text{num}}$ governs only the alignment weight, while $\tau_{\text{denom}}$ governs only the dispersion sharpening: the two temperatures decouple into separate terms in the $m \to \infty$ limit. This is the formal statement behind the decomposition informally stated in Section 3.

E.5.2. CATEGORICAL ANCHORING: THE ALIGNMENT TERM DECOUPLES OVER NUMERICAL FEATURES

**Theorem 6** (Categorical Anchoring). *Partition the feature indices $[n] = \mathcal{N}_{num} \sqcup \mathcal{N}_{cat}$. Assume the SCARF corruption operator sets $p_{corr}(j) = 0$ for every $j \in \mathcal{N}_{cat}$ and $p_{corr}(j) = p_{corr} \in (0, 1]$ for every $j \in \mathcal{N}_{num}$, with independent coin flips across features. Let $\tilde{x}_j$ denote $x$ with only the $j$-th feature resampled from its marginal. Then the alignment term of the asymptotic loss (6) satisfies*

$$\mathbb{E}_{(x,\tilde{x}) \sim p_{pos}} \big[ \|\phi(x) - \phi(\tilde{x})\|^2 \big] = p_{corr} \sum_{j \in \mathcal{N}_{num}} \mathbb{E}_x \big[ \|\phi(x) - \phi(\tilde{x}_j)\|^2 \big] + O(p_{corr}^2). \tag{9}$$

*In particular, the first-order alignment loss is independent of how $\phi$ acts on the categorical coordinates: no encoder variation along the categorical subspace is penalized at leading order in $p_{corr}$.*

*Proof.* Let $C_j \in \{0, 1\}$ be the Bernoulli($p_{corr}(j)$) coin indicating whether feature $j$ is corrupted (so $C_j = 0$ with certainty when $j \in \mathcal{N}_{cat}$). By the SCARF operator, $\tilde{x}$ agrees with $x$ on every index where $C_j = 0$ and is resampled from the marginal where $C_j = 1$. Since the $\{C_j\}$ are independent,

$$\mathbb{P}(\text{exactly one feature corrupted}) = \sum_{j \in \mathcal{N}_{num}} p_{corr}(1 - p_{corr})^{|\mathcal{N}_{num}|-1},$$

$$\mathbb{P}(\text{no feature corrupted}) = (1 - p_{corr})^{|\mathcal{N}_{num}|}, \qquad \mathbb{P}(\text{two or more corrupted}) = O(p_{corr}^2).$$

Conditional on the "no corruption" event, $\tilde{x} = x$ and $\|\phi(x) - \phi(\tilde{x})\|^2 = 0$. Conditional on "exactly feature $j$ is corrupted," $\tilde{x} = \tilde{x}_j$ by construction, so $\|\phi(x) - \phi(\tilde{x})\|^2 = \|\phi(x) - \phi(\tilde{x}_j)\|^2$. The $O(p_{corr}^2)$ term absorbs the two-or-more-corruption events, which have probability $O(p_{corr}^2)$ and contribute a term bounded by $4 \cdot O(p_{corr}^2)$ since $\|\phi(x) - \phi(y)\|^2 \le 4$ on the unit sphere. Taking total expectation,

$$\mathbb{E}\|\phi(x) - \phi(\tilde{x})\|^2 = 0 \cdot (1 - p_{corr})^{|\mathcal{N}_{num}|} + p_{corr}(1 - p_{corr})^{|\mathcal{N}_{num}|-1} \sum_{j \in \mathcal{N}_{num}} \mathbb{E}\|\phi(x) - \phi(\tilde{x}_j)\|^2 + O(p_{corr}^2).$$

Expanding $(1 - p_{corr})^{|\mathcal{N}_{num}|-1} = 1 + O(p_{corr})$ and absorbing the cross-term into the $O(p_{corr}^2)$ remainder yields (9).

Because no term in the sum on the right-hand side involves a categorical index $j \in \mathcal{N}_{cat}$, the alignment penalty is unchanged under any replacement $\phi \to \phi'$ that agrees with $\phi$ on all numerical-feature variation but differs on the categorical subspace. Thus the alignment term imposes no leading-order pressure on the encoder to suppress categorical features. $\square$

*Remark* 7. Theorem 6 is the formal version of the statement at the start of this appendix "the alignment term places no penalty on categorical features." Combined with Proposition 2, it says: Robinson's suppression mechanism (which requires $z_S$ constant across *both* positives and negatives) does not apply to the categorical-anchored SCARF setup, because negatives are sampled i.i.d. from $p_{data}$ and therefore exhibit variation in $z_S$. The dispersion term then favors encoders that distinguish categorical classes (since pushing distinct categorical values to disjoint regions lowers the log-mean-exp in (6)), but a formal guarantee of this second step requires non-degenerate density assumptions on the categorical marginal and is beyond the present scope; we note it as an empirical observation (see Section 3, Fig. 2).

E.5.3. CONSISTENCY OF WEIGHTEDRISK

The next theorem formalizes the intuition at the end of Section 3: when $k$ most-similar historical cases are all labeled non-erroneous, the posterior error probability at the query is very low. We prove this as a bias/variance decomposition of a Nadaraya/Watson-type estimator, tailored to our vendor-scoped retrieval setup.

**Theorem 8** (WeightedRisk Consistency). *Fix a vendor $v$ and suppose historical transactions $\{x_j\}_{j=1}^N$ are i.i.d. $\sim p_v$, each with binary error label $Y(x_j) \in \{0, 1\}$ drawn independently given $x_j$ from $p_{err}(x_j) := \mathbb{P}(Y = 1 \mid x = x_j)$. Assume:*

- (i) **Lipschitz posterior.** $|p_{err}(x) - p_{err}(x')| \le L\|\phi(x) - \phi(x')\|$ for some $L > 0$ and all $x, x'$ in the support of $p_v$.

- (ii) **Bounded kernel.** There exists $s_{\min} \in (0, 1]$ such that the similarity weights used by WeightedRisk satisfy $s_{\min} \le w_j := \text{sim}\big(\phi(x_q), \phi(x_j)\big) \le 1$ for every $x_j \in \mathcal{N}_k(x_q)$. (This holds under the paper's threshold $\theta$: the k-NN is restricted to the $\theta$-similar subset.)

*(iii) **Query in support.** $x_q$ lies in the interior of the support of $p_v$ (so every open ball around $\phi(x_q)$ has positive $p_v$-measure).*

*Then, letting $\mathcal{N}_k(x_q)$ denote the $k$ nearest neighbors of $x_q$ among the historical set under the embedding metric $d(x, x') = \|\phi(x) - \phi(x')\|$, the WeightedRisk estimator*

$$\widehat{R}_k(x_q) := \frac{\sum_{x_j \in \mathcal{N}_k(x_q)} w_j \, Y(x_j)}{\sum_{x_j \in \mathcal{N}_k(x_q)} w_j}$$

*satisfies:*

1. *(Finite-$k$ bias.)* $\left| \mathbb{E}[\widehat{R}_k(x_q)] - p_{err}(x_q) \right| \leq L \cdot \mathbb{E}\left[ \max_{x_j \in \mathcal{N}_k(x_q)} \|\phi(x_j) - \phi(x_q)\| \right].$

2. *(Consistency.)* *If $k = k(N) \to \infty$ and $k(N)/N \to 0$ as $N \to \infty$, then $\widehat{R}_k(x_q) \xrightarrow{p} p_{err}(x_q)$.*

*Proof.* Denote $W := \sum_j w_j$ and $\alpha_j := w_j/W$, so $\alpha_j \in [0, 1]$ and $\sum_j \alpha_j = 1$. By assumption (ii), $W \geq k s_{\min}$.

**Step 1: Bias bound.** Conditional on the set $\mathcal{N}_k(x_q)$ and the points $\{x_j\}_{x_j \in \mathcal{N}_k(x_q)}$ (but before $\{Y(x_j)\}$ are drawn), the $Y(x_j)$ are independent with $\mathbb{E}[Y(x_j) \mid x_j] = p_{\mathrm{err}}(x_j)$. Hence

$$\mathbb{E}\left[\widehat{R}_k(x_q) \,\big|\, \mathcal{N}_k(x_q)\right] = \sum_{x_j \in \mathcal{N}_k(x_q)} \alpha_j \, p_{\mathrm{err}}(x_j).$$

Subtracting $p_{\mathrm{err}}(x_q) = \sum_j \alpha_j p_{\mathrm{err}}(x_q)$ (using $\sum_j \alpha_j = 1$) and applying the triangle inequality plus Lipschitz assumption (i):

$$\left| \mathbb{E}[\widehat{R}_k(x_q) \mid \mathcal{N}_k] - p_{\mathrm{err}}(x_q) \right| \leq \sum_j \alpha_j \, |p_{\mathrm{err}}(x_j) - p_{\mathrm{err}}(x_q)| \leq L \sum_j \alpha_j \|\phi(x_j) - \phi(x_q)\| \leq L \max_j \|\phi(x_j) - \phi(x_q)\|.$$

Taking expectation over $\mathcal{N}_k(x_q)$ and applying the law of total expectation yields the bias bound claimed in (1).

**Step 2: Variance bound.** Conditional on $\mathcal{N}_k(x_q)$, $\widehat{R}_k(x_q) = \sum_j \alpha_j Y(x_j)$ is a convex combination of $k$ independent $\{0, 1\}$-bounded random variables. By Hoeffding's inequality (Hoeffding, 1963), for every $t > 0$,

$$\mathbb{P}\left( \left| \widehat{R}_k(x_q) - \mathbb{E}[\widehat{R}_k \mid \mathcal{N}_k] \right| > t \,\Big|\, \mathcal{N}_k \right) \leq 2 \exp\left( -\frac{2t^2}{\sum_j \alpha_j^2} \right).$$

Since $\alpha_j \leq 1/(k s_{\min})$ and $\sum_j \alpha_j^2 \leq (\max_j \alpha_j) \cdot \sum_j \alpha_j = \max_j \alpha_j \leq 1/(k s_{\min})$, the exponent is at most $-2t^2 k s_{\min}$, yielding

$$\mathbb{P}\left( \left| \widehat{R}_k(x_q) - \mathbb{E}[\widehat{R}_k \mid \mathcal{N}_k] \right| > t \,\Big|\, \mathcal{N}_k \right) \leq 2 \exp(-2t^2 k s_{\min}). \tag{10}$$

This bound is uniform in $\mathcal{N}_k$; unconditioning gives the same bound on the unconditional probability.

**Step 3: $k$-NN distance vanishes (Lemma below).** We now show that under assumption (iii) and $k/N \to 0$,

$$D_k := \max_{x_j \in \mathcal{N}_k(x_q)} \|\phi(x_j) - \phi(x_q)\| \xrightarrow{p} 0.$$

For any $r > 0$, let $q_r := \mathbb{P}_{x \sim p_v}(\|\phi(x) - \phi(x_q)\| \leq r)$. By (iii), $q_r > 0$. The number of samples $x_j$ within $r$ of $x_q$ is $M_r \sim \mathrm{Binomial}(N, q_r)$, with $\mathbb{E}[M_r] = N q_r$ and $\mathrm{Var}(M_r) = N q_r (1 - q_r)$. By Chebyshev's inequality,

$$\mathbb{P}(M_r < k) \leq \mathbb{P}\left( |M_r - N q_r| > N q_r - k \right) \leq \frac{N q_r (1 - q_r)}{(N q_r - k)^2}.$$

Under $k/N \to 0$, we have $k/(N q_r) \to 0$ for every fixed $r > 0$, so $N q_r - k \geq \frac{1}{2} N q_r$ eventually, and the right-hand side is at most $4(1 - q_r)/(N q_r) \to 0$. Thus $\mathbb{P}(M_r \geq k) \to 1$, and on that event $D_k \leq r$. Since $r$ was arbitrary, $D_k \xrightarrow{p} 0$.

**Step 4: Combine.** By the triangle inequality,

$$\left|\widehat{R}_k(x_q) - p_{\text{err}}(x_q)\right| \leq \left|\widehat{R}_k(x_q) - \mathbb{E}[\widehat{R}_k \mid \mathcal{N}_k]\right| + \left|\mathbb{E}[\widehat{R}_k \mid \mathcal{N}_k] - p_{\text{err}}(x_q)\right|.$$

Fix $\varepsilon > 0$. The first term exceeds $\varepsilon/2$ with probability at most $2\exp(-\varepsilon^2 k s_{\min}/2)$ by (10) with $t = \varepsilon/2$; this vanishes as $k \to \infty$. The second term is bounded by $L \cdot D_k$ which $\xrightarrow{p} 0$ by Step 3, so it exceeds $\varepsilon/2$ with vanishing probability. A union bound gives $\mathbb{P}(|\widehat{R}_k - p_{\text{err}}(x_q)| > \varepsilon) \to 0$, proving consistency. $\qquad\square$

*Remark* 9. The BinaryRisk variant $\widehat{R}_k^{\text{bin}}(x_q) := \mathbb{1}[\exists x_j \in \mathcal{N}_k(x_q) : Y(x_j) = 1]$ admits a simpler bound: under assumption (i),

$$\mathbb{P}\left(\widehat{R}_k^{\text{bin}}(x_q) = 0 \mid \mathcal{N}_k(x_q)\right) = \prod_j (1 - p_{\text{err}}(x_j)) \leq \exp\left(-\sum_j p_{\text{err}}(x_j)\right) \leq \exp\left(-k\left(p_{\text{err}}(x_q) - LD_k\right)\right),$$

so the false-negative probability of BinaryRisk decays exponentially in $k$ with rate $p_{\text{err}}(x_q) - LD_k$. This formalizes the remark at Section 3 that "the probability of individual investigative errors is multiplied $k$ times."

## F. Two-Stage Pipeline: TabPFN with SCARF for False Positive Reduction

This section presents a two-stage pipeline for fraud detection that combines transformer-based probabilistic forecasting (TabPFN) with contrastive representation learning (SCARF). The approach addresses a core challenge in anomaly detection: achieving high precision without sacrificing recall.

### F.1. Pipeline Architecture

Our proposed pipeline consists of two distinct stages:

1. **High-Recall Detection Stage**: A TabPFN (Transformer-based Probabilistic Forecasting Network) classifier trained to identify potential anomalies with high recall

2. **False Positive Reduction Stage**: A SCARF (Self-supervised Contrastive learning with Augmented Reality Features) model that learns representations that distinguish between true positives and false positives

This decoupled architecture offers several advantages: (1) it allows each model to specialize in a single task, (2) it provides transparency and interpretability at each stage, and (3) it offers flexibility in deployment, as stages can be optimized independently.

### F.2. Stage 1: High-Recall Anomaly Detection with TabPFN

TabPFN is a transformer-based model designed for tabular data that employs a Bayesian formulation for classification. We selected TabPFN for the first stage because of its ability to achieve high recall on imbalanced datasets without requiring extensive hyperparameter tuning.

We trained TabPFN on a subset of the Santander Transaction dataset, which contains approximately 10% anomalies. To address class imbalance, we employed a specialized sampling strategy that retains all anomalies while subsampling normal transactions. This approach ensures the model is exposed to all anomaly patterns while maintaining a manageable dataset size.

As predicted, TabPFN achieved high recall (0.92) at the expense of precision (0.20), resulting in a model that captured most anomalies but generated many false positives. This behavior aligns with our goal for the first stage, as we prioritize capturing true anomalies even at the cost of false alarms.

### F.3. Stage 2: False Positive Reduction with SCARF

The second stage addresses the primary limitation of the TabPFN model: the high number of false positives. For this purpose, we employ SCARF, a self-supervised contrastive learning framework that learns robust representations by contrasting corrupted and uncorrupted instances.

Unlike traditional supervised approaches, SCARF does not rely directly on labels but instead learns to differentiate between semantically similar and dissimilar instances through contrastive loss. We adapt SCARF specifically for false positive reduction by:

1. Focusing exclusively on TabPFN's positive predictions

2. Using ground truth labels to differentiate between true positives (actual anomalies) and false positives (misclassified normal transactions)

3. Implementing a curriculum learning strategy with gradually decreasing temperature and corruption rate

The SCARF model architecture consists of an encoder network that maps input features to embedding space. During training, for each sample $x$, SCARF generates a corrupted version $\tilde{x}$ by randomly perturbing a subset of features. The model is trained to bring embeddings of original and corrupted versions closer while pushing embeddings of different samples apart.

$$\mathcal{L}_{\text{contrastive}} = -\log \frac{\exp(\text{sim}(z_i, \tilde{z}_i)/\tau)}{\sum_{j=1}^{N} \mathbb{1}_{[j \neq i]} \exp(\text{sim}(z_i, z_j)/\tau)} \tag{11}$$

where $z_i$ represents the embedding of original sample $x_i$, $\tilde{z}_i$ represents the embedding of corrupted version $\tilde{x}_i$, $\text{sim}(u, v)$ is the cosine similarity, and $\tau$ is the temperature parameter.

Our curriculum learning strategy gradually reduces both the temperature parameter and corruption rate according to a predefined schedule, allowing the model to learn increasingly fine-grained distinctions between similar instances:

*Table 7.* **Curriculum schedule:** 50-epoch alternating adjustments of $\tau_{\text{denom}}$ and $p_{\text{corr}}$ used by SimGuard.

| Stage | Temperature | Corruption Rate |
|:-----:|:-----------:|:---------------:|
| 1 | 1.0 | 0.5 |
| 2 | 0.8 | 0.5 |
| 3 | 0.8 | 0.4 |
| 4 | 0.5 | 0.4 |
| 5 | 0.5 | 0.3 |
| 6 | 0.2 | 0.3 |
| 7 | 0.2 | 0.2 |
| 8 | 0.1 | 0.2 |
| 9 | 0.05 | 0.2 |

After training the SCARF encoder on TabPFN's positive predictions, we use a $k$-nearest-neighbors classifier on the learned embeddings to distinguish true positives from false positives. Headline numbers for this two-stage pipeline appear in Table 4.

## G. Extended Conclusion, Limitations and Future Work

SimGuard's context-aware retrieval depends on a sufficiently populated historical repository. Two regimes where it degrades: (i) entirely novel fraud patterns where no similar historical cases exist (cold start), and (ii) extremely sparse data regions lacking reliable $k$-NN neighbors. Future directions include hybrid approaches that combine SimGuard with traditional anomaly detectors for cold-start handling, feature-specific corruption rates based on importance scores, and incorporation of temporal transaction patterns to capture seasonal variations and evolving vendor relationships. The asymmetric-temperature-decoupling principle extends to other heterogeneous-tabular domains such as healthcare diagnostics and manufacturing quality control.

**Deployment pattern.** SimGuard pairs naturally with a high-recall first-stage classifier: TabPFN flags at ∼0.92 recall and SimGuard then suppresses the 13% to 31% of flagged cases that resemble known benign patterns, yielding a precision/recall trade-off curve operators can tune to business risk tolerance.

