# OpenReview forum: "SimGuard: Context-Aware Anomaly Filtering via Similarity-Guided Error Detection"
_ICML.cc/2026/Workshop/FMSD — FMSD @ ICML 2026 Poster_

### Official Review · Reviewer_7e8p · 2026-05-18
**a novel guarding method for anomaly reduction in financial domain**

**Rating:** 7
**Confidence:** 4

**Review:**

The paper proposes a guarding method via asymetrical InforNCE loss, with curriculum learning for tuning \tau-denom (denominator temperature in the InforNCE loss), p-corr (feature corrpution probability) and a retrieval mechanism during inference for financial transactions. Extensive experiments support and justify the motivation of the proposed method.

---

### Official Review · Reviewer_mofK · 2026-05-19
**Interesting post-hoc filtering setup, but the methodology and claims need clearer validation**

**Rating:** 7
**Confidence:** 4

**Review:**

### Summary

This paper proposes SimGuard, a post-hoc filtering framework for anomaly-flagged invoices. The method combines a contrastive tabular encoder, asymmetric temperature scaling, curriculum scheduling, and vendor-scoped nearest-neighbor retrieval over historical cases. The practical setting is relevant, and the paper studies a meaningful false-positive reduction problem. However, several central claims of the paper, including its generality and its self-supervised nature, would benefit from clearer validation and explanation.

### Strengths

1. The paper studies a practically relevant problem: reducing false positives in a high-recall anomaly pipeline without repeatedly retraining the system.

2. The use of historical cases as a post-hoc filtering mechanism is operationally reasonable in settings where similar false alarms may recur over time.

3. The enterprise experiments suggest that the proposed pipeline may reduce review volume while largely preserving recall, which is the most relevant practical objective in this setting. The experiments are in general comprehensive.

4. The paper includes both empirical and theoretical components, and the overall structure of the method is relatively easy to follow.

### Areas for Improvement

1. The paper is presented as a general post-hoc filtering methodology for foundation-model classifiers, but the actual two-stage pipeline appears to be built specifically around TabPFN. The paper does not study how sensitive the method is to the choice of upstream detector.

2. The claim that the method is self-supervised is somewhat confusing. While the representation learning stage is self-supervised, the inference-stage retrieval explicitly depends on historical labeled cases, so the full pipeline is not label-free.

3. The theoretical results look potentially relevant, but at first glance they seem fairly close to re-applications or extensions of known arguments. The paper would be stronger if it clarified more explicitly what is genuinely new in the proofs and what specific insight the theory provides for the proposed detection/filtering setting.

### Detailed Comments

1. My main concern is about generality. The paper is written in a way that suggests a broadly applicable post-hoc filtering methodology for foundation-model classifiers, but the empirical pipeline is effectively instantiated with TabPFN. If the methodology is intended to be general, it would be important to test whether similar behavior holds with other upstream tabular classifiers.

2. I was also somewhat confused by the self-supervised claim. My understanding is that the encoder is trained in a self-supervised manner, but the actual filtering decision at inference time depends on historical investigator-labeled cases retrieved from the repository. If that reading is correct, then the paper should distinguish more carefully between self-supervised representation learning and the full end-to-end system.

3. The theory is an interesting part of the paper, but its role is not yet fully clear. The asymptotic loss decomposition and the consistency result for the weighted retrieval score seem reasonable, but the paper should explain more explicitly how these proofs differ from prior analyses and what specific new theoretical insight is gained for this problem.

### Justification of Score

Overall, I think the paper studies a relevant problem and contains some promising ideas. My main reservations are that the paper’s claims about generality and self-supervision are not yet fully supported by the current validation, and that the theoretical contribution would benefit from clearer positioning relative to prior work.

---

### Official Review · Reviewer_G3d8 · 2026-05-21

**Rating:** 7
**Confidence:** 3

**Review:**

### Summary

The paper proposes SimGuard, a post-hoc filtering layer placed downstream of high-recall tabular classifiers (e.g., TabPFN) to suppress false positives. It combines three ideas: an asymmetric InfoNCE loss with decoupled alignment/dispersion temperatures, a 50-epoch curriculum that jointly schedules the dispersion temperature and SCARF-style corruption rate, and vendor-scoped few-shot retrieval that scores flagged cases via similarity-weighted voting over investigator-verified history. On two enterprise invoice datasets (450K / 800K), SimGuard cuts flagged volume by 15% with a recall drop below 1%, and also improves F1 over TabPFN on Santander and Credit Card Fraud.


### Strengths

- The ‘high-recall classifier + post-hoc filter’ decomposition directly matches real enterprise auditing pipelines. The three contributions also cleanly map to three research questions in the experiments.

- Decomposing the contrastive objective into alignment and dispersion terms gives the asymmetric-temperature choice a theoretical justification rather than a pure heuristic.

-  Ablations show a monotonic gain from joint scheduling of temperature and corruption rate over fixed parameters or corruption-only. This supports the claim that the curriculum, not any single component, drives performance.

- Vendor-scoped retrieval with retraining-free pattern absorption and case-level explanations are valuable for human-in-the-loop auditing, and clearly differentiate SimGuard from black-box classifiers.


### Weaknesses

- F1 improvements over TabPFN on Santander and Credit Card Fraud are small in absolute terms, and XGBoost still outperforms SimGuard on Credit Card Fraud. Multi-seed variance or a significance test would strengthen these claims.

- The retrieval relies on a fixed neighbor count and similarity threshold, but no sensitivity sweep is provided in the main paper.

- The retrieval layer falls back to flagging when historical neighbors are insufficient, but the paper does not report how often this triggers or its impact on new vendors with sparse history.